# Antiviral Mechanism of Tea Polyphenols against Porcine Reproductive and Respiratory Syndrome Virus

**DOI:** 10.3390/pathogens10020202

**Published:** 2021-02-13

**Authors:** Xun Wang, Wenjuan Dong, Xiaoxiao Zhang, Zhenbang Zhu, Yaosheng Chen, Xiaohong Liu, Chunhe Guo

**Affiliations:** State Key Laboratory of Biocontrol, School of Life Sciences, Sun Yat-sen University, North Third Road, Guangzhou Higher Education Mega Center, Guangzhou 510006, Guangdong, China; wangx597@mail2.sysu.edu.cn (X.W.); dongwj5@mail2.sysu.edu.cn (W.D.); zhangxx55@mail2.sysu.edu.cn (X.Z.); 007583@yzu.edu.cn (Z.Z.); chyaosh@mail.sysu.edu.cn (Y.C.); liuxh8@mail.sysu.edu.cn (X.L.)

**Keywords:** PRRSV, tea polyphenol, antiviral, cytokine

## Abstract

Neither inactivated nor attenuated vaccines can effectively prevent and control the infection and spread of porcine reproductive and respiratory syndrome virus (PRRSV). Therefore, it is necessary to broaden new horizons and to conceive effective preventive strategies. The main components of Tea polyphenol (TPP) are catechins and their derivatives. TPP has many physiological activities and has certain antiviral and antifungal effects. However, whether TPP shows anti-PRRSV activity remains unclear. We found that TPP effectively inhibited PRRSV infection in Marc-145 cells by suppressing the stages of viral attachment, internalization, replication, and release. TPP exhibited a potent anti-PRRSV effect regardless of pre-treatment or post-treatment. In addition, we demonstrated that TPP restrained PRRSV-induced p65 entry into the nucleus to suppress the activation of the NF-κB signaling pathway, which ultimately leads to the inhibition of the expression of inflammatory cytokines. Furthermore, TPP limited the synthesis of viral non-structural protein 2 (nsp2), the core component of viral replication transcription complexes, which may contribute to the inhibition of viral RNA replication. TPP has the potential to develop into an effective antiviral agent for PRRSV prevention and control in the future.

## 1. Introduction

Porcine reproductive and respiratory syndrome virus (PRRSV) is one of the most important pathogens that continuously impacts the swine industry worldwide. PRRSV was first recognized in the late 1980s in North America and Europe [1,2]. It belongs to the order *Nidovirales*, family *Arteriviridae*, and is a small, enveloped virus (diameter about 65 nm) containing a single-stranded RNA genome of positive polarity. Its genome is about 15 kb in length, which contains at least 11 open reading frames [3]. The virus mainly grows in porcine alveolar macrophages and causes acute pneumonia and reproductive and respiratory problems in pigs [4,5,6]. 

According to the current knowledge, PRRSV mutates rapidly at an estimated rate of 3.29 × 10^−3^ substitutions per nucleotide per year, developing rapidly growing evolutionary strains [7,8]. Due to its high antigenic variability and poorly understood immunopathogenesis, there is currently no effective vaccine or treatment to control PRRSV infection [9]. 

According to the theory of traditional Chinese medicine, more and more natural ingredients have been proved to have the functions of disease prevention and health care [10,11,12]. Tea polyphenol (TPP) is the general term of polyphenols in tea leaves. The main components of TPP are catechins and their derivatives [13]. TPP has many physiological activities, such as anti-oxidation, anti-radiation, anti-aging, blood lipid lowering, blood sugar lowering, and bacteria and enzyme inhibition [14,15]. It is a compound with polyphenolic structural properties, such as catechins and anthocyanins [16]. TPP is the main component of green tea soup, which has certain antiviral and antifungal effects [17,18,19]. Epigallocatechin gallate (EGCG), which accounts for 60% to 80% of TPP, has been reported for its antiviral effect on several viruses such as hepatitis C virus, chikungunya virus, hepatitis B virus, and Zika virus [20,21,22,23,24].

However, whether TPP has an inhibitory effect on PRRSV infection and replication remains unknown. Here, we demonstrated that TPP potently inhibited PRRSV infection in Marc-145 cells in a dose-dependent manner. The mechanism of PRRSV inhibition by TPP was also investigated. TPP inhibited the attachment, internalization, replication, and release stages of the PRRSV life cycle. TPP could inhibit p65 transport into the nucleus, thus suppressing the activation of the NF-κB signaling pathway, which ultimately results in the inhibition of the expression of inflammatory cytokines induced by PRRSV infection. In addition, TPP was capable of blocking the synthesis of viral non-structural protein 2 (nsp2), the core component of replication transcription complexes (RTC), which then leads to the suppression of the translation and assembly of viral proteins. In our study, TPP is an effective drug against PRRSV infection and has a broad application value in the swine industry.

## 2. Results

### 2.1. TPP Can Inhibit the Replication of PRRSV 

To identify the antiviral activity of TPP against PRRSV, we first used the alamarBlue^®^ assay to test the cytotoxicity of TPP in Marc-145 cells. As shown in Figure 1A, with increasing concentrations of TPP, the cell viability rate (%) was not affected, and TPP at a concentration of no higher than 100 μg/mL showed no cytotoxic effect. Next, we examined the anti-PRRSV effect of TPP by immunofluorescence microscopy and qRT-PCR at 36 h post-infection (hpi). As shown in Figure 1B,C, PRRSV was significantly inhibited by TPP in a dose-dependent manner. The green fluorescence measurement by immunofluorescence assay (IFA) imaging significantly reduced. We further tested the effect of TPP on PRRSV infection at different time points. As shown in Figure 1D,E, treatment of cells with TPP resulted in a significant reduction of the mRNA and protein levels of PRRSV N. We further used PRRSV strain CHR6 at different multiplicity of infection (MOI) to infect Marc-145 cells. TPP still exhibited strong anti-PRRSV activity (Figure 1F,G). Due to the high mutation rate of PRRSV, we analyzed the effect of TPP on different virus strains. As shown in Figure 1H, treatment with TPP showed a powerful inhibitory effect on these strains. These results indicate that TPP has potent inhibition against PRRSV infection. 

### 2.2. Pre-Treatment and Post-Treatment of TPP Show a Potent Inhibitory Effect on PRRSV Infection

Since TPP plays a powerful role in inhibiting PRRSV (Figure 1), we next treated Marc-145 cells with TPP before or after PRRSV infection to further explore the influence of TPP on PRRSV. The results showed that PRRSV was effectively inhibited when cells were pre-treated with TPP for 2 h and then infected with PRRSV for 24 h (Figure 2A,C). In the post-treatment assay, TPP also showed a potent inhibitory effect on the mRNA and protein levels of PRRSV N (Figure 2B,D). These data demonstrate that TPP restrains PRRSV replication regardless of its pre-treatment or post-treatment. 

### 2.3. TPP Blocks Viral Attachment, Internalization, Replication, and Release

Since both pre-treatment and post-treatment of TPP play an effective inhibitory effect on virus replication, we then explored which stage(s) of viral infection was/were interrupted by TPP treatment. To investigate this, we designed viral attachment, entry, replication, and release assays, as described in Figure 3A. For virus binding, Marc-145 cells were infected with PRRSV-CHR6 (MOI = 6) in the presence or absence of TPP for 4 h at 4 °C, which allows virus binding but not internalization (Figure 3A, treatment B), and then collected. As shown in Figure 3B, TPP treatment showed an inhibitory effect on PRRSV binding to Marc-145 cells. To examine whether TPP may also affect the internalization of PRRSV, virus-infected Marc-145 cells were treated with TPP for 2 or 4 h (Figure 3A, treatment C). As shown in Figure 3C, the viral N protein was significantly inhibited by TPP, suggesting that TPP also inhibits PRRSV internalization.

For replication, Marc-145 cells were infected with PRRSV for 8 h at 37 °C to realize normal virus replication. The infected cells were then treated with TPP for 2 or 4 h (Figure 3A, treatment D), and cells were collected. As shown in Figure 3D, TPP treatment significantly reduced the viral N protein level, suggesting that TPP inhibits the replication stage of PRRSV. We further examined whether TPP could affect PRRSV release (Figure 3A, treatment E). We measured the viral N protein level in the supernatant, and we saw that the expression of N protein was significantly reduced after TPP treatment. TPP had potent effects on the release phase of PRRSV infection (Figure 3E).

### 2.4. TPP Treatment Reduces the Expression of p65 and Impairs p65 Transport into the Nucleus after PRRSV Infection

To investigate whether the NF-κB signal pathway is affected by TPP, the location of NF-κB p65 was tested in Marc-145 cells treated with TPP. As shown in Figure 4A, TPP significantly inhibited the mRNA expression of NF-κB p65 induced by PRRSV infection and LPS (L2630, Sigma, USA) stimulation (positive control). PRRSV infection led to the translocation of p65 into the cell nucleus, resulting in the activation of the NF-κB pathway. However, upon TPP treatment, red fluorescence representing p65 located in the nucleus was drastically reduced in both virus-infected and LPS-treated cells (Figure 4B). These data show that TPP inhibits p65 transport into the nucleus, which is caused by PRRSV infection, thus inhibiting the activation of the NF-κB signaling pathway.

### 2.5. TPP Treatment Decreases Cytokine Expression Induced by PRRSV in Marc-145 Cells

Since TPP inhibits the activation of the NF-κB signaling pathway, we speculated that TPP could limit PRRSV-induced cytokine expression. To demonstrate the hypothesis, we explored the effect of TPP on the expression of cytokines such as interferon-β (IFN-β), interleukin-6 (IL-6), IL-8, and tumor necrosis factor-α (TNF-α), which are known to be related to the host antiviral and inflammatory reactions. Upon TPP treatment, the mRNA levels of IFN-β, IL-6, IL-8, and TNF-α significantly diminished in both PRRSV-infected and LPS-stimulated Marc-145 cells (positive control) (Figure 5A–D). Compared to the mock-treated cells, the cytokine expression displayed a comparable level in cells treated with TPP alone. These data suggest that TPP may restrain PRRSV replication via inhibiting virus-induced expression of cytokines.

### 2.6. TPP Inhibits the Synthesis of PRRSV nsp2, the Core Component of Viral RTC

Since TPP effectively inhibits the replication of PRRSV, we speculated that the inhibition may be attributed to its effect on the assembly of replication transcription complexes, in which viral nsp2 plays a crucial role. To validate the hypothesis, we tested the expression of viral nsp2 upon TPP treatment of Marc-145 cells. As shown in Figure 6A, compared to the cells with PRRSV treatment alone, green fluorescence representing nsp2 expression remarkably reduced in the presence of TPP in Marc-145 cells. To further demonstrate the inhibition of viral nsp2 by TPP treatment, Marc-145 cells were transfected with the mCherry-tagged nsp2 expression plasmid or an empty vector for 6 h and then mock-treated or treated with TPP for another 24 h. As shown in Figure 6B, compared to untreated cells, TPP-treated cells showed a significant inhibiting effect on viral nsp2 expression, with no influence on mCherry empty vector expression, which suggests that TPP directly inhibits the synthesis of PRRSV nsp2.

## 3. Discussion

PRRSV has spread rapidly all over the world, which has lasted for many years. In recent years, the prevention and control of the virus have become more and more complex, the diversity of virus strains has been increasing, and new virus strains are emerging. Because of its great variability and persistent infection, PRRSV is difficult to control [25,26,27]. Moreover, due to the abuse of vaccines, many newly emerged PRRSV infections are caused not by wild-type strains but by vaccine viruses [28,29]. In recent years, there have been some new vaccines with specific adjuvants, but they have little protective effect. Some drugs, such as herbal extracts, compounds, siRNA, microRNA, and neutralizing antibodies, have been shown to inhibit PRRSV replication in vitro [30,31,32]. However, their antiviral persistence is not clear, and it is far from being applied to the actual pig industry.

In our study, TPP inhibits the replication of PRRSV in multiple ways. Likewise, other polyphenols have also been described to present antiviral activity, such as proanthocyanidin A2 and theaflavin [33,34]. Previous reports also indicate that replication of PRRSV in Marc-145 cells is inhibited by EGCG [24], which accounts for 60% to 80% of TPP. However, TPP is the total content of polyphenols in tea, showing better antiviral properties.

We conclude TPP has multiple potential mechanisms of viral inhibition, as described in Figure 7. On the one hand, TPP blocks the attachment and internalization of PRRSV or inhibits the assembly of viral RTC after virions enter cells during the virus life cycle. TPP also shows strong inhibition of PRRSV mature particle release into the extracellular environment. On the other hand, TPP is capable of restraining PRRSV-induced translocation of NF-κB p65 into the nucleus, thereby suppressing the expression of cytokines, which may contribute to its inhibition of PRRSV [35]. In addition, many signal transduction pathways can be modulated by TPP to control pro-inflammatory gene expression, such as JAK-STAT, TLR, and PI3K-AKT [36]. More studies are required to verify whether TPP has any influence on these signal pathways.

From the effects of pre-treatment and post-treatment of TPP on PRRSV replication, the effect of the pre-treatment approach seems to be better. The middle region of viral nsp2 is highly heterogeneous and responsible for size variation among PRRSV strains [37,38]. Variations might contribute to viral fitness by regulating viral mRNA synthesis, suggesting that viral nsp2 is a critical component of viral RTC. In addition, assembly of PRRSV RTC requires a network of viral nsps, including nsp2 [39,40]. In this study, we showed that TPP inhibits the synthesis of nsp2, thereby blocking the formation of viral RTC. However, the mechanism of TPP underlying the inhibition of viral RTC formation remains unclear and needs further study.

Due to the high mutation rate in the PRRSV genome, we explored the TPP antiviral effects on different strains, and our results showed that TPP can effectively inhibit these strains’ infections. VR-2332 is a prototypical North-American-type isolate; TJM-92 and JXA1 are highly pathogenic strains; CHR6, isolated in China, is a classic North-American-type strain. Because PRRSV nsp2 is a highly mutable protein, we performed gene comparison in nsp2 among the above strains and showed that these strains display 66.7% identity in nsp2 (data not shown). Although these strains show a huge difference in nsp2, our results verify that the suppression of PRRSV by TPP is broad spectrum and not strain dependent. 

In addition to antiviral functions, TPP also has a powerful influence on microbiota growth and modifies the composition of the gut microbiota to improve immune responses and decrease inflammatory responses [12]. TPP has strong anti-oxidant and pro-oxidant activities, which promotes the growth of strictly anaerobic bacteria, while suppressing many opportunistic pathogens [41]. TPP could selectively inhibit the growth of or kill a wide range of Gram-positive and Gram-negative bacteria by destroying their cell wall and cell membrane permeability [42,43,44,45]. Supplementation of TPP to feed promotes the growth of *Bifidobacterium* and *Lactobacillus* in calves [46], and this is associated with decreased incidences of digestive and respiratory diseases [47]. The beneficial effects of TPP in the elimination of pathogenic and other deleterious bacteria are obvious. PRRSV mainly infects and destroys porcine alveolar macrophages and leads to severe immunosuppression, which may promote infection by *Mycoplasma pneumoniae*, *Streptococcus*, and other pathogens [48]. Further studies are needed to elucidate the role of the microbiota in mediating the effects of TPP on PRRSV infection.

Furthermore, TPP can exert its significant anti-inflammatory properties by regulating the activation or deactivation of inflammation-related or oxidative-stress-related cell signaling pathways, such as NF-κB, MAPK, Nrf2, and STAT1/3 pathways [49], which may be hijacked by PRRSV [50,51,52]. Whether TPP affects the above pathways except for NF-κB to inhibit PRRSV replication is unknown. Further studies are needed to address this.

## 4. Materials and Methods

### 4.1. Cells and Viruses

Marc-145 cells (China Center for Type Culture Collection, Wuhan, China), an immortalized cell line derived from African green monkey kidney cells, were cultured in Dulbecco’s modified Eagle’s medium (DMEM) (Corning, Phoenix, AZ, USA) containing 10% fetal bovine serum (FBS) (PAN, Aidenbach, Germany), which are permissive to PRRSV replication and are commonly used in laboratories. PRRSV strain CHR6 (classical North-American-type PRRSV strain) was provided by Dr. Guihong Zhang from South China Agricultural University, and PRRSV-EGFP, a recombinant virus showing growth replication characteristics similar to those of the wild-type virus in the infected cells, was gifted by Dr. Shuqi Xiao from Northwest A&F University. Other three PRRSV strains, TJM-92 (GenBank accession no. EU860248.1), JXA1 (GenBank accession no. EF112445.1), and VR-2332 (GenBank accession no. EF536003.1), were also used in the present study. The above PRRSV strains were used to infect Marc-145 cells. The virus strains were propagated in Marc-145 cells and titrated as a 50% tissue culture infective dose (TCID_50_).

### 4.2. Preparation of TPP

TPP, with a purity of 99%, was gifted by Guangzhou Zhongnongda Co. LTD, Guangzhou, China. TPP mainly consists of catechins and flavonoids and was dissolved in sterile water. TPP was checked for no potential contamination.

### 4.3. Cytotoxicity Assay

The cytotoxicity of TPP was detected with the alamarBlue^®^ assay (Invitrogen, Carlsbad, CA, USA) according to the manufacturer’s instructions. Marc-145 cells (1 × 10^4^ cells/well) were seeded in 96-well plates, and different concentrations of TPP were added in DMEM when cells grew to 60–70% confluence. After incubation of Marc-145 cells for 48 h, 10 μL of alamarBlue^®^ was added to each well, and the cells were incubated for another 3 h. Finally, the fluorescence value was detected using the Multi-Mode Reader (Synergy2, BioTek, Winooski, VT, USA) at an absorbance of 570 nm.

### 4.4. Quantitative Real-Time Reverse Transcription Polymerase Chain Reaction (qRT-PCR)

To detect the relative expression of PRRSV ORF7 and cytokines, qRT-PCR was performed. Marc-145 cells (2 × 10^6^ cells/well) were seeded in six-well plates, and RNA was extracted from cultured cells using TRIzol reagent (Magen, Guangzhou, China). The Reverse Transcription System (A3500, Promega, Madison, WI, USA) was used for reverse transcription in 20 μL of reaction volume following the manufacturer’s instructions. The reverse transcription primers were oligo (dT) 15 primer (C110A, Promega, Madison, WI, USA) and random primer (C118A, Promega, Madison, WI, USA). Reverse transcription products were amplified by the LightCycler 480 Real-Time PCR System (LC480, Roche, Basel, Switzerland) using 2× RealStar Green Power Mixture (GenStar, Guangzhou, China). The primers are listed in Table 1. The qRT-PCR reaction system was run under the following conditions: 95 °C for 10 min, then 95 °C for 15 s, 60 °C for 1 min and 72 °C for 30 s through 40 cycles, and finally 72 °C for 10 min. Data were normalized to GAPDH in each individual sample. The 2^−ΔΔCt^ method was used to calculate relative expression changes. Relative expression (fold changes) was compared to mock-infected cells.

### 4.5. Western Blot

Six-well-plate cell samples (2 × 10^6^ cells/well) were harvested in cell lysis buffer (Beyotime, Shanghai, China) containing PMSF (Beyotime, Shanghai, China). Processed samples were subjected to 12% sodium dodecyl sulfate–polyacrylamide gel electrophoresis (SDS-PAGE) and transferred onto a polyvinyl difluoride (PVDF) membrane (Millipore, Boston, MA, USA). Then the membranes were blocked with 5% BSA (Ruishu, China) in TBST (20 Mm Tris-HCl pH 8.0, 150 mM NaCl, 0.05% Tween 20) for 1 h at 37 °C. After blocking, they were incubated with an anti-PRRSV N protein monoclonal antibody (1:1000 dilution, Jeno Biotech, Inc., Chuncheon-si, South Korea) and an anti-glyceraldehyde phosphate dehydrogenase (GAPDH) antibody (1:1000 dilution, Cell Signaling Technology, Danvers, MA, USA) overnight at 4 °C. After washing three times with TBST, membranes were incubated with an anti-mouse IgG, HRP-linked antibody or anti-rabbit IgG, HRP-linked antibody (1:1000 dilution, Cell Signaling Technology, Danvers, MA, USA) for 1 h at 37 °C. The antibody signals were exposed using an enhanced chemiluminescence (ECL) reagent (Fdbio Science, Guangzhou, China). 

### 4.6. Antiviral Assay

Cells were seeded in six-well plates (2 × 10^6^ cells/well) and grown to 70–80% confluence. There were two approaches to analyze the antiviral effect of TPP. 

(I) Pre-treatment: Cells were pre-treated with different concentrations of TPP (0, 50, and 100 μg/mL) for 2 h, and PRRSV-CHR6 was then added and cultured for 36 h. 

(II) Post-treatment: Cells were inoculated with PRRSV-CHR6 for 4 h, then the inoculum was removed, and TPP (0, 50, and 100 μg/mL) was added to the cells for 36 h.

### 4.7. Viral Attachment, Entry, and Replication Assays

Marc-145 cells (2 × 10^6^ cells/well) were seeded in six-well plates. For the attachment assay, cells were cooled for 2 h at 4 °C and then infected with PRRSV-CHR6 at a MOI of 6 in the presence of different concentrations of TPP (0, 25, and 50 μg/mL) for 4 h at 4 °C. Then the cells were collected for Western blot analysis so that we could determine the effect of TPP on viral attachment. As for the entry assay, cells were inoculated with PRRSV-CHR6 (MOI = 6) for 4 h at 4 °C. After binding to the cell surface, cells were washed with PBS three times and cultured at 37 °C for 2 or 4 h in the presence of various concentrations of TPP (0, 25, and 50 μg/mL). The cells were collected for Western blot analysis so that we could determine the effect of TPP on viral internalization. As for the replication assay, cells were inoculated with PRRSV-CHR6 (MOI = 6) at 37 °C for 8 h, and then various concentrations of TPP (0, 25, and 50 μg/mL) were added for 2 or 4 h. The cells were collected for Western blot analysis so that we could determine the effect of TPP on viral replication. 

### 4.8. Release Assays

Marc-145 cells (2 × 10^6^ cells/well) were seeded in six-well plates. For the release assay, cells were incubated with PRRSV-CHR6 (MOI = 5) for 12 h at 37 °C. After that, cells were rinsed with PBS three times and TPP at different concentrations was added to the cells for 3 h at 37 °C. Finally, cells were collected for Western blot analysis to detect the PRRSV N protein expression. To detect the release of the viral N protein into the supernatant, 1 mL of conditioned medium of each group was collected and incubated with 20% trichloroacetic acid for 1 h. The mixture was then centrifuged for 15 min at 4 °C. The supernatant was discarded, and the pellet was washed with ice-cold acetone to remove trichloroacetic acid. The acetone was removed by centrifugation. After drying at room temperature for 15 min, the protein pellet was dissolved in 40 μL of sample buffer and heated at 95 °C for 15 min. The entire sample was loaded for Western blot.

### 4.9. Immunofluorescence Assay (IFA)

Marc-145 cells (8 × 10^5^ cells/well) were seeded in 12-well plates or confocal Petri dishes. Cells were fixed in 4% paraformaldehyde for 15 min. After permeabilization with 0.5% Triton X-100 for 15 min at room temperature (RT), cells were blocked with 1% bovine serum albumin (BSA) for 30 min and then incubated with a rabbit monoclonal antibody against the p65 protein (1:500 dilution, Cell Signaling Technology) or an antibody against PRRSV nsp2 (a gift from Dr. Hanchun Yang, China Agricultural University, 1:1000) at 4 °C overnight. Cells were then incubated with an anti-rabbit secondary antibody conjugated with Alexa Fluor® 555 or 488 (Cell Signaling Technology, MA, USA) at 1:1000 dilution for 2 h. Nuclei were stained using DAPI (1:1000; Cell Signaling Technology). Cells were examined by fluorescence microscopy (Nikon, Tokyo, Japan).

### 4.10. PRRSV nsp2 Inhibition Assay

The PRRSV nsp2 gene was amplified from the PRRSV-CHR6 strain and cloned into vector pmCherry-N (632523, Takara, Kusatsu, Japan) with a C-terminal mCherry tag. Marc-145 cells (2 × 10^6^ cells/well) were seeded in six-well plates. The mCherry-tagged nsp2 plasmid was transfected into Marc-145 cells for 6 h, and cells were then mock-treated or treated with TPP at concentrations of 50 and 100 μg/mL. After incubation for 24 h, cells were collected for Western blot analysis to detect the inhibitory effect of TPP on nsp2 expression.

### 4.11. Statistical Analysis

All experiments were performed with at least three independent replicates. Student’s *t*-test and one-way ANOVA were used to analyze the data. Statistical analysis was performed using SPSS 17.0 and GraphPad Prism 6.0. *p* < 0.05 was considered to be significant.

## 5. Conclusions

In conclusion, our study demonstrates that TPP shows a potent antiviral effect on PRRSV infection by targeting the attachment process of the viral life cycle and down-regulating virus-induced inflammatory cytokines in infected cells. These results indicate that TPP has the potential to develop into a novel drug to prevent and control PRRSV infection in the future.

## Figures and Tables

**Figure 1 pathogens-10-00202-f001:**
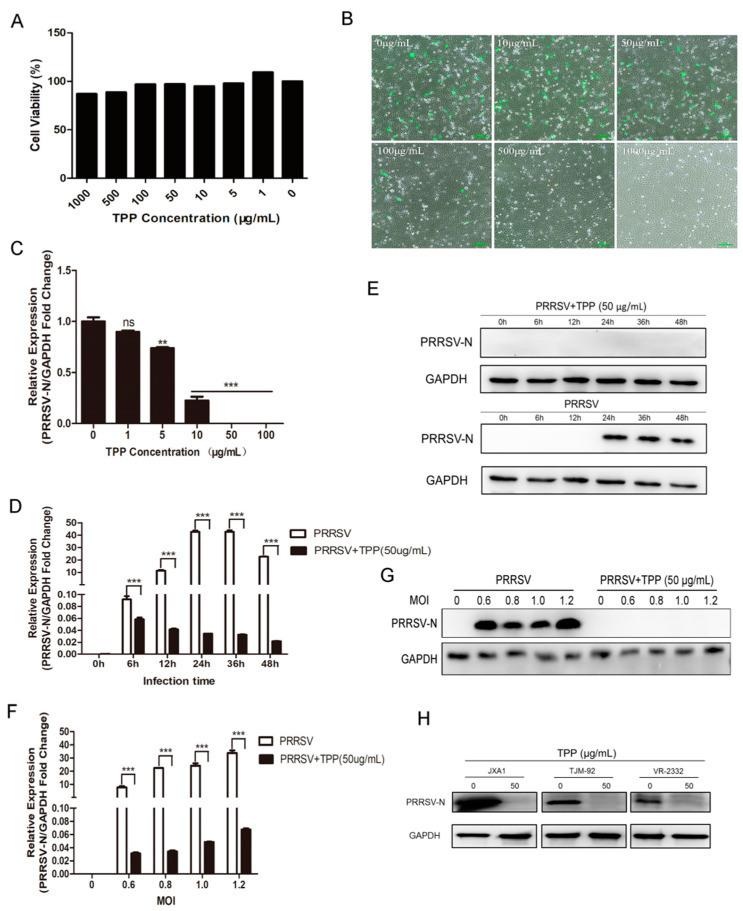
Tea polyphenol (TPP) restrains the infection and replication of porcine reproductive and respiratory syndrome virus (PRRSV) in Marc-145 cells. (**A**) The cytotoxicity of TPP was measured by the alamarBlue^®^ assay. Marc-145 cells were treated with TPP at indicated concentrations for 48 h, and a cell viability assay was performed. (**B**) Marc-145 cells were infected with PRRSV-EGFP (MOI = 0.6) in the presence of different concentrations of TPP for 36 h and then were harvested for fluorescence microscope examination. Scale bar, 100 μm. (**C**) Marc-145 cells were infected with PRRSV-CHR6 (MOI = 0.6) in the presence of different concentrations of TPP for 36 h. The mRNA expression of viral ORF7 (PRRSV N) was detected by qRT-PCR. (**D**,**E**) Marc-145 cells were infected with PRRSV-CHR6 (MOI = 0.6) in the presence or absence of TPP for different time points, the mRNA level of viral ORF7 (PRRSV N) was detected by qRT-PCR (**D**), and PRRSV N protein was determined by Western blot (**E**). (**F**,**G**) Marc-145 cells were infected with PRRSV-CHR6 at different MOIs in the presence or absence of TPP for 36 h, and the expression of viral ORF7 (PRRSV N) was detected by qRT-PCR (F) and N protein was determined by Western blot (**G**). (**H**) Marc-145 cells were infected with JXA1, TJM-92, and VR-2332 strains (MOI = 0.6) in the presence or absence of TPP for 36 h, and the viral N protein was determined by Western blot. Data are representative of the results of three independent experiments (means ± SE). Significant differences compared to the control group are denoted by ** *p* < 0.01, and *** *p* < 0.001.

**Figure 2 pathogens-10-00202-f002:**
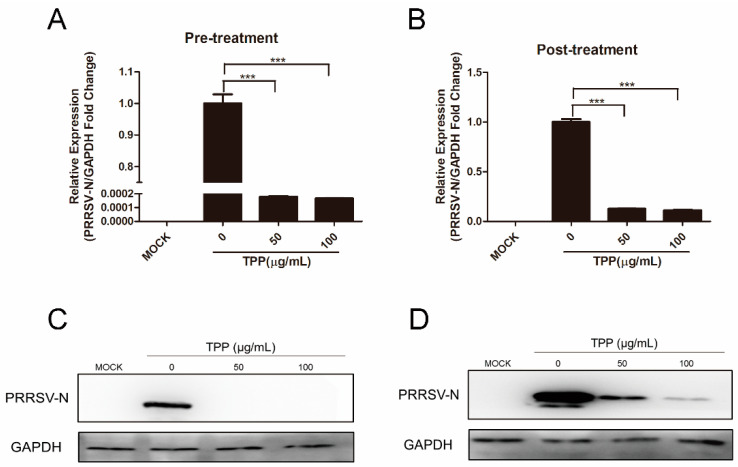
TPP suppresses PRRSV infection regardless of its pre-treatment and post-treatment. (**A**,**C**) Marc-145 cells were pre-treated with different concentrations of TPP for 2 h and then infected with PRRSV-CHR6 for 36 h, and the expression of viral ORF7 (PRRSV N) was detected by qRT-PCR (**A**) and viral N protein was determined by Western blot analysis (**C**). (**B**,**D**) Marc-145 cells were infected with PRRSV-CHR6 (MOI = 0.6) for 4 h, then TPP was added, cells were incubated for another 36 h, and the expression of viral ORF7 (PRRSV N) was detected by qRT-PCR (B) and N protein was determined by Western blot (**D**). Data are representative of the results of three independent experiments (means ± SE). Significant differences compared to the control group are denoted by *** *p* < 0.001.

**Figure 3 pathogens-10-00202-f003:**
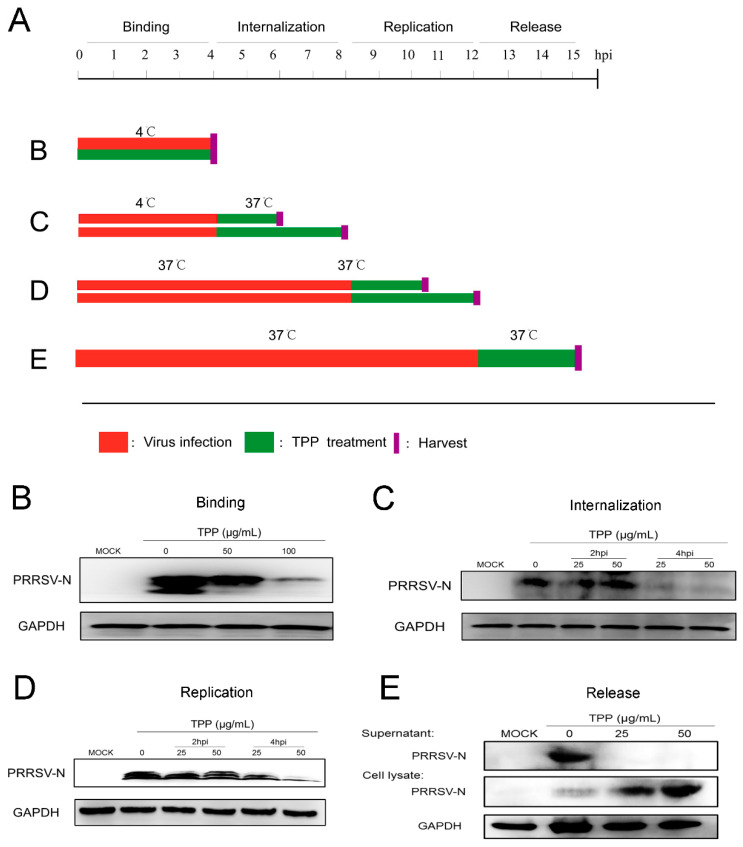
TPP inhibits PRRSV attachment, entry, replication, and release. Marc-145 cells were infected with PRRSV-CHR6 at an MOI of 6, and the infected cells were cultured in the presence of various concentrations of TPP and collected at indicated time points post-infection to determine viral N protein levels by Western blot. (**A**) Different TPP treatment schemes. Red bars represent the PRRSV infection period, green bars represent TPP treatment, magenta vertical bars represent removing TPP, and purple vertical bars represent cell harvest. (**B**–**E**) Viral binding, internalization, replication, and release were performed in cells treated, as described in (**A**), and then cells were harvested for Western blot analysis. Data are representative of the results of three independent experiments.

**Figure 4 pathogens-10-00202-f004:**
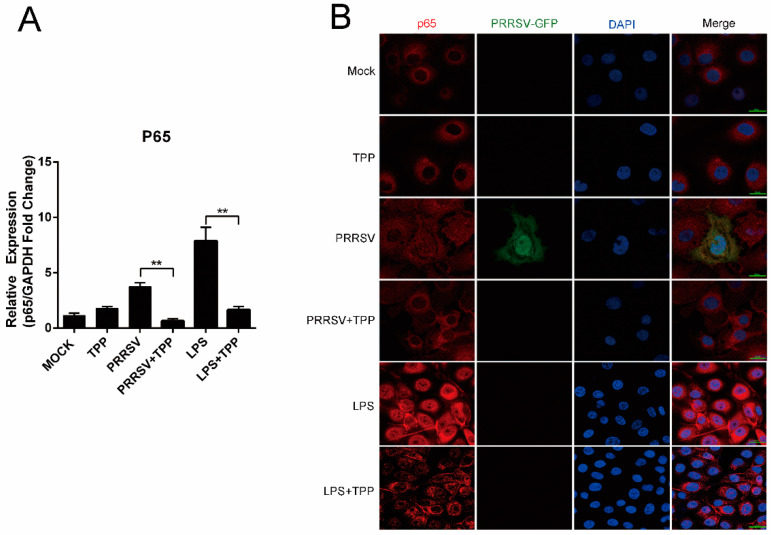
TPP inhibits the NF-κB signaling pathway in PRRSV-infected Marc-145 cells. (**A**) Marc-145 cells were infected with PRRSV-CHR6 (MOI = 0.6) in the presence or absence of TPP, and the mRNA level of p65 was assessed by qRT-PCR. (**B**) Marc-145 cells were infected with PRRSV-EGFP (MOI = 0.6) in the presence or absence of TPP, and immunofluorescence assay (IFA) for the p65 protein was performed at 36 h post-infection (hpi) using Alexa-Fluor-555-conjugated anti-rabbit secondary antibody (red). Nuclei were stained with DAPI (blue). Scale bar, 20 μm. LPS served as a positive control. Data are representative of the results of three independent experiments (means ± SE). Significant differences compared to the control group are denoted by ** *p* < 0.01.

**Figure 5 pathogens-10-00202-f005:**
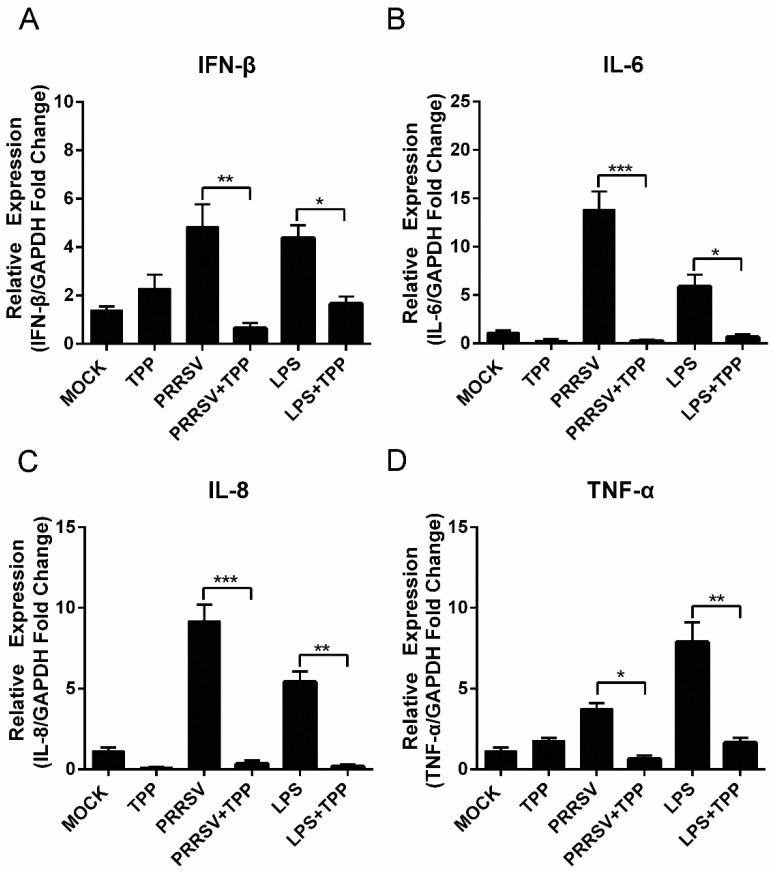
Expression of cytokines in Marc-145 cells treated with TPP. (**A**–**D**) Marc-145 cells were mock-infected or infected with PRRSV-CHR6 (MOI = 0.6) in the presence or absence of TPP (50 μg/mL) for 12 h. The expression of pro-inflammatory cytokines IFN-β (**A**), IL-6 (**B**), IL-8 (**C**), and TNF-α (**D**) was analyzed using qRT-PCR. Relative expression (fold) in comparison with mock-infected cells (set up as 1) is shown. LPS served as a positive control. Data are the results of three independent experiments (means ± SE). Significant differences between PRRSV-infected cells and those treated with TPP are denoted by * *p* < 0.05, ** *p* < 0.01, and *** *p* < 0.001.

**Figure 6 pathogens-10-00202-f006:**
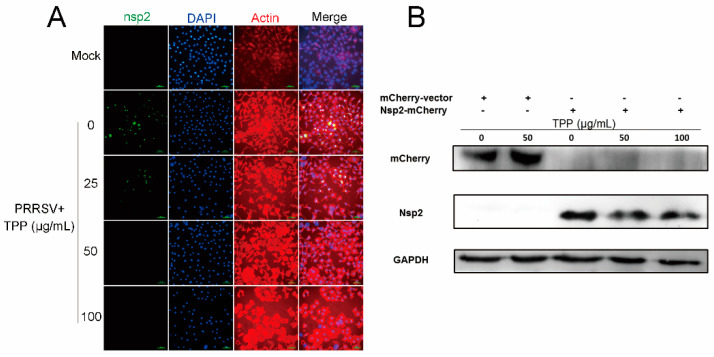
TPP inhibits the synthesis of viral nsp2 in infected Marc-145 cells. (**A**) Marc-145 cells were infected with PRRSV-CHR6 (MOI = 0.6) for 6 h, then different concentrations of TPP were added, and cells were incubated for another 36 h. IFA for the nsp2 protein of PRRSV was performed using Alexa-Fluor-488-conjugated anti-mouse secondary antibody (green). The actin protein was stained with Alexa-Fluor-555-conjugated anti-rabbit secondary antibody (red). Nuclei were stained with DAPI (blue). Scale bar, 100 μm. (**B**) The mCherry-tagged nsp2 plasmid was transfected into Marc-145 cells for 6 h, and cells were then mock-treated or treated with TPP for 24 h. Cells were finally harvested for Western blot analysis. Data are representative of the results of three independent experiments.

**Figure 7 pathogens-10-00202-f007:**
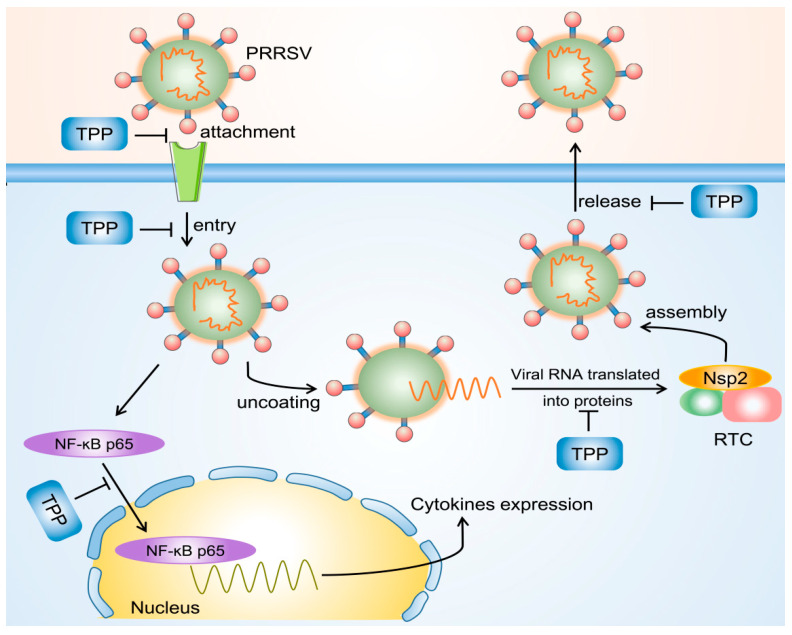
The schematic diagram of PRRSV inhibition by TPP. On the one hand, TPP blocks the attachment, internalization, and release of PRRSV or inhibits the assembly of viral RTC after virions enter cells during the virus life cycle. On the other hand, TPP is capable of restraining PRRSV-induced translocation of NF-κB p65 into the nucleus, thereby suppressing the expression of cytokines, which may contribute to its inhibition of PRRSV.

**Table 1 pathogens-10-00202-t001:** List of the primers used in this study.

Primer ^a^	Sequence (5′–3′) ^b^
**N-F**	AAAACCAGTCCAGAGGCAAG
**N-R**	CGGATCAGACGCACAGTATG
**GAPDH-F**	TGACAACAGCCTCAAGATCG
**GAPDH-R**	GTCTTCTGGGTGGCAGTGAT
**P65-F**	AGAGCCTCCTGCACCAGTTCT
**P65-R**	-R TCACTCCTTCTTCCTG
**IFN-β-F**	GCAATTGAATGGAAGGCTTGA
**IFN-β-R**	CAGCGTCCTCCTTCTGGAACT
**IL-6-F**	AGAGGCACTGGCAGAAAAC
**IL-6-R**	TGCAGGAACTGGATCAGGAC
**IL-8-F**	CACTGTGAAAATTCAGAAATCATTGTTA
**IL-8-R**	CTTCACAAATACCTGCACAACCTTC
**TNF-α-F**	TCTGTCTGCTGCACTTTGGAGTGA
**TNF-α-R**	TTGAGGGTTTGCTACAACATGGGC

^a^ F, forward primer; R, reverse primer. ^b^ Green monkey gene sequences and PRRSV gene sequences were downloaded from GenBank.

## Data Availability

All data analyzed during this study are included in this published article. The raw data are available from the corresponding author upon reasonable request.

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
