# Peer review of "Antiviral Mechanism of Tea Polyphenols against Porcine Reproductive and Respiratory Syndrome Virus"

_pathogens, 2021, doi:10.3390/pathogens10020202_

Round 1

Reviewer 1 Report

The Manuscript Titled “Antiviral mechanism of tea polyphenols on porcine reproductive and respiratory syndrome virus” investigates Tea polyphenols (TPP) and their derivatives anti-porcine reproductive and respiratory syndrome virus (PRRSV) activity. The major finding of the study is that TPP effectively inhibited PRRSV infection in Marc-145 cells by suppressing the stages of viral attachment, internalization, replication and release. The study also outlines the potential for TPP as an effective antiviral agent for PRRSV prevention and control in the future. This is an important field of study based on the overall burden of the disease. The manuscript has been written very well and is an important contribution to the field.

Recommendation: Approval with minor revisions

Minor Comments

  1. Under materials and methods please include information about how many Marc-145 cells (for example: 1 million/200 ul) were cultured for the proposed experiments.
  2. Please include information if the TPP was checked for potential contamination? If yes, include that information in the materials and methods section.

Author Response

The following is a point-to-point response to the reviewers’ comments.

Reviewer reports:
Reviewer 1: The Manuscript Titled “Antiviral mechanism of tea polyphenols on porcine reproductive and respiratory syndrome virus” investigates Tea polyphenols (TPP) and their derivatives anti-porcine reproductive and respiratory syndrome virus (PRRSV) activity. The major finding of the study is that TPP effectively inhibited PRRSV infection in Marc-145 cells by suppressing the stages of viral attachment, internalization, replication and release. The study also outlines the potential for TPP as an effective antiviral agent for PRRSV prevention and control in the future. This is an important field of study based on the overall burden of the disease. The manuscript has been written very well and is an important contribution to the field.

Minor Comments:

1. Under materials and methods please include information about how many Marc-145 cells (for example: 1 million/200 ul) were cultured for the proposed experiments.

Thank you for your insightful comments of our paper. According to your suggestion, we have added the information in our study in lines 280, 310, 317, 331, 343-344, and 355-356.

2. Please include information if the TPP was checked for potential contamination? If yes, include that information in the materials and methods section.

We have checked the TPP purity, and affirmed that it has no potential contamination, we have added it in line 269.

Reviewer 2 Report

I think this is a well thought out study and provides an interesting conclusion. The findings will need to be investigated in porcine directly to understand the role of the immune response on the progression of the infection. The question I have as an outside reviewer is the following:

1) What happens to the results if NF-Kn is knocked down (if possible), would TPP have any effect on other pathways? Does the virus use NF-KB signaling pathway only? This can be addressed in your current model

2) The authors specify the high mutation rate of the virus, do these results hold true with different strain of the virus? 

3) Is there another Chinese medicine that does or does not have these effects?
